# Correlation of breast cancer microcirculation construction with tumor stem cells (CSCs) and epithelial-mesenchymal transition (EMT) based on contrast-enhanced ultrasound (CEUS)

**Xiaoling Leng[1], Guofu Huang[2], Siyi Li[1], Miaomiao Yao[1], Jianbing Ding📷[3]\*, Fucheng Ma[1]\***

**1** Department of Ultrasound, the Affiliated Tumor Hospital of Xinjiang Medical University, Urumqi, Xinjiang, China, **2** Department of Hematology and Oncology, the Fifth Affiliated Hospital of Xinjiang Medical University, Urumqi, Xinjiang, China, **3** Basic Medical College, Xinjiang Medical University, Urumqi, Xinjiang, China

\* dingjb1234@aliyun.com, 1601379937@qq.com (JD); mafuchengchao@sina.com (FM)

## Abstract

### Objective

This study is to explore the correlation between the contrast-enhanced ultrasound (CEUS) characteristics of breast cancer and the epithelial-mesenchyme transformation (EMT).

### Methods

Totally 119 patients of breast cancer underwent CEUS. Tissues in the active area were collected and subjected to the immunohistochemical detection, PT-PCR and Western blot. Correlation analysis was conducted between the clinical pathological parameters and the CEUS indicators.

### Results

The expression levels of CD44, N-cadherin, and β-catenin in breast cancer tissues were higher than those in adjacent tissues (P<0.05). However, the expression levels of CD24 and E-cadherin in breast cancer tissues were lower than those in adjacent tissues (P<0.05). There was no significant difference in *E-cadherin* mRNA and Vimentin levels between cancer and adjacent tissues (P>0.05). The expressions were up-regulated in the CSCs, with higher histological grade, lymph node metastasis, and negative estrogen receptor (ER) expression. Smaller breast tumors, with no lymph node metastasis, lower clinical stage, and positive ER expression, tended to exhibit the up-regulated epithelial phenotype. Breast tumors, with high histological grade, lymph node metastasis, high clinical staging grade, and negative ER expression, tended to exhibit the up-regulated interstitial phenotype. The peak intensity of the time-intensity curve (TIC) for the CEUS was positively correlated with the CSC marker CD44 and the interstitial phenotype marker N-cadherin. The starting time of enhancement was negatively correlated with the N-cadherin. Area under the curve was

**Data Availability Statement:** All relevant data are within the manuscript. Please check Fig 1–5 and Tables 1–4 for details.

**Funding:** This work was supported by the project of scientific and technological assistance to Xinjiang (No.2020E0269).

**Competing interests:** The authors have declared that no competing interests exist.

positively correlated with the expression of CD44 and N-cadherin, while negatively correlated with the epithelial phenotype marker β-catenin. The time to peak was negatively correlated with the interstitial phenotypes Vimentin and N-cadherin, with no correlation with the E-cadherin or β-catenin.

## Conclusion

Breast cancers show the enlarged lesions after enlargement and perfusion defect for the CEUS. The fast-in pattern, high enhancement, and high perfusion in the TIC are correlated with the CSCs and EMT expressions, suggesting poor disease prognosis.

## Introduction

Breast cancer accounts for 23% of the malignant tumors, and early screening and prognosis assessment is of great significance for the breast cancers [1]. At present, the puncture biopsy has been commonly used in the diagnosis of breast cancer. However, due to the tumor tissue heterogeneity, the pathological results of the punctured tumor tissue might be limited [2], and the puncture biopsy is usually invasive. Therefore, non-invasive assessments for breast are needed for the prognosis of breast cancers. The relationship between the ultrasound features and the biological features of breast cancers has been quantitatively evaluated based on the radiomics, and the accuracy of predicting the hormone receptor expression in breast cancer is as high as 67.7%, indicating that the tumor features could be obtained at the genetic and cellular levels through the ultrasound images [3].

Contrast-enhanced ultrasound (CEUS) can display the microvascular structure within the tumor in real time, with a high spatial and temporal resolution. CEUS can visually exhibit the characteristics of the microcirculation within the tumor, which can be quantitatively analyzed besides the qualitative evaluation [4]. In our previous study, the qualitative CEUS indicators well correlated with the prognostic factors (i.e., the tumor size, histological grade, and clinical stage) have been screened and selected, including the perfusion defect, enhancement range, enhancement degree, and time to peak, peak intensity, starting time of enhancement, and area under the curve [5].

Cancer stem cells (CSCs) and epithelial-mesenchymal transition (EMT) are key regulators for the breast cancer aggressiveness [6]. Breast cancer CSCs have the self-renewal and multi-directional differentiation abilities, which are closely related to the biological behavior of tumorigenesis, proliferation, metastasis, and drug resistance. The EMT refers to the loss of tight junctions between cells, loss of polarity, and gaining the mesenchymal cell characteristics, accompanied by the up-regulated N-cadherin expression and down-regulated E-cadherin expression, which has been shown to be related to the tumor invasion, metastasis, and treatment resistance [6]. CSCs and EMT cells are closely related with each other, which share similar characteristics.

It has been shown that CD44 and VEGF are co-expressed in cancer tissues [7], suggesting synergistic effects on tumor microangiogenesis and metastasis. The occurrence of EMT in the CSCs would promote the proliferation and migration of fibroblasts and induce angiogenesis, also supporting the association between the CSCs and breast cancer microvessels. CEUS based on microcirculation perfusion can not only display the tumor blood vessels, but also provide the evidence concerning the microcirculation hemodynamics, which might contribute to the investigation of the aggressiveness of breast cancer [8]. However, the extent to which the

CEUS characteristics of breast cancer can reflect the distribution of CSCs and the EMT extension is still unclear.

In this study, the macroscopic imaging-based CEUS technology was used to reflect the tumor microcirculation perfusion for the imaging analysis of breast cancer lesions. More active cancerous tissues were obtained through puncture under the real-time guidance of CEUS. The presence of CSCs and the extension of EMT in these tissues were analyzed, and its correlation with CEUS characteristics was also studied, in order to screen for the CEUS indicators that better reflect breast cancer aggressiveness and prognosis.

## Materials and methods

### Study subjects

A total of 119 females with breast cancer who were admitted to our hospital from January 2017 to October 2018 were included in this study, with a median age of 46 years. Inclusion criteria were as follows: patients meeting the diagnostic criteria for breast cancer, who had received CEUS, without mental disorders. Exclusion criteria included: patients with pregnancy; with moderate to severe anemia; in menstrual periods; with vaginal bleeding of unknown reason; or having previously received radiation, chemotherapy, and/or surgical treatment. All patients underwent CEUS examination before puncture. The specimens were collected from the most representative plane under the guidance of CEUS. The obtained tissue specimens were subjected to immunohistochemical examination, PT-PCR and Western blot analysis. The study was approved by the Ethics Committee of the Tumor Hospital Affiliated to the Xinjiang Medical University, and the written informed consent was obtained from each patient.

### CEUS analysis

The CEUS analysis was performed with the Logic E9 color Doppler ultrasound system (GE, Boston, USA), with a 9L-4 probe and the mechanical index of 0.16. For detection, the patient was lying on his back, and the morphology, size, and edge of the tumor was checked with the 2D ultrasound, with the probe position maintained. Totally 5 ml contrast agent was quickly injected into the elbow vein of the healthy upper limb, followed by 5 ml 0.9% sodium chloride injection. After the contrast agent injection, the dynamic images were recorded for 2 min 30 s, until the enhanced lesion image was reduced. The qualitative indicators of contrast agent perfusion were analyzed, including whether the lesion range was enlarged after enhancement compared with the 2D ultrasound, and whether there was perfusion defect within the lesion. The regions of interest (ROIs) covered the entire lesion area, and the normal breast tissue of the same depth was used as the control area, to obtain the time intensity curve (TIC). The indicators were observed, including the peak intensity, time to peak, starting time of enhancement, and area under the curve (Fig 1). The time to peak was the time between the start of CEUS enhancement and the peak intensity within the ROIs. The peak intensity was the maximum enhanced intensity of the ROIs. The area under the curve was defined as total volume of contrast medium (or blood) traversing the region of interest. The representative sections of CEUS were marked before surgery.

### Immunohistochemistry

After surgery, the lesion was dissected and sliced, in consistency of the CEUS sections. The immunohistochemistry was performed with the Envision two-step method, to detect the CSCs markers (i.e., the CD44 and CD24) and the EMT markers (i.e., the E-cadherin, β-catenin, vimentin, N-cadherin). The following primary antibodies were used herein: rabbit anti-E

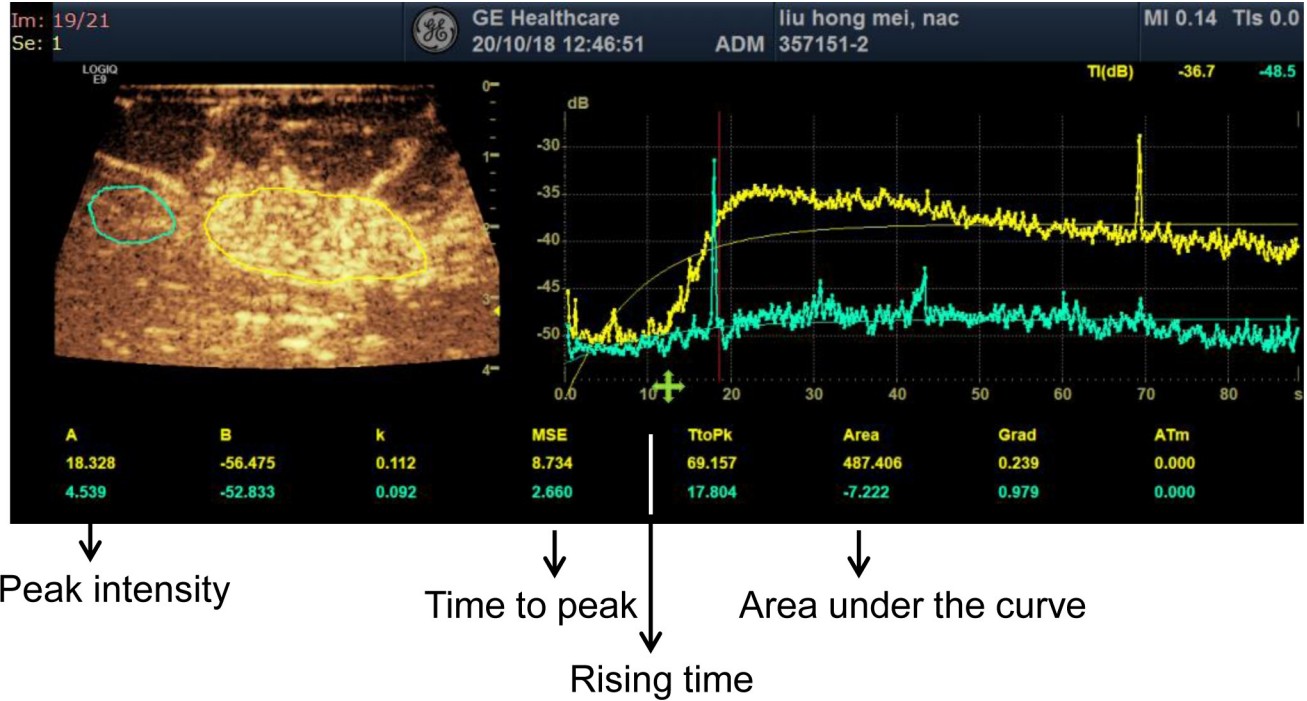

**Fig 1. Time intensity curve analysis of contrast-enhanced ultrasound.** The peak intensity, time to peak, rising time, and, the area under the curve were indicated in the figure.

Cadherin monoclonal antibody (1:100 dilution; ab40772; Abcam, Cambridge, MA, USA), rabbit anti-beta Catenin monoclonal antibody (1:100 dilution; ab32572; Abcam), mouse anti-Vimentin monoclonal antibody (1:100 dilution; ab8978; Abcam), rabbit anti-N Cadherin monoclonal antibody (1:100 dilution; ab76011; Abcam), rabbit anti-CD44 polyclonal antibody (1:100 dilution; ab157107; Abcam), and mouse anti-CD24 monoclonal antibody (1:100 dilution; MA5-11833; Invitrogen, Carlsbad, CA, USA). Criteria for immunohistochemical results were as follows: CD44 and E-cadherin were positively expressed in the cell membrane and cytoplasm of tumor cells; CD24 was positively expressed in the cytoplasm of tumor cells; N-cadherin and β-catenin were positively expressed in the cell membrane of tumor cells; Vimentin was positively expressed in the cytoplasm of the tissue around the cancer nest. The scoring criteria were as follows: 5 high-power fields were randomly observed in each section, and 100 tumor cells were counted. The staining results were evaluated by the Fromowitz comprehensive scoring method. The percentage of positive tumor cells was used to obtain a semi-quantitative score, indicating the relative intensity of protein expression. According to the Sinicrope modified method, the percentage of positive tumor cells was calculated as follows: < 5%, 0; 5%-25%, 1; 25%-50%, 2; 50%-75%, 3; and >75%, 4.

## Quantitative real-time PCR

Totally 100 mg breast cancer tissue was ground with liquid nitrogen, and total RNA was extracted with TRLZOL. After determining the concentrations, the cDNA was obtained with the revere transcription PCR. Quantitative real-time PCR was performed with the SYBR Green Select Mix on the PCR machine. Primer sequences were as follows: E-cadherin, forward 5′-CGAGAGCTACACGTTCACGG-3′ and reverse 5′-GGGTGTCGAGGGAAAAATAGG-3′; β-catenin, forward 5′-AGCTTCCAGACACGCTATCAT-3′ and 5′-CGGTACAACGAGCTGTTTCTAC-3′; and

Vimentin, forward 5′-AGTCCACTGAGTACCGGAGAC-3′ and 5′-CA TTTCACGCATCTGG CGTTC-3′; N-cadherin, forward 5′-AGCCAACCTTAACTGAGGAGT-3′ and reverse 5′-GG CAAGTTGATTGGAGGGATG-3′; CD44, forward 5′-CTGCCGCTTTGCAGGTGTA-3′ and reverse 5′-CATTGTGGGCAAGGTGCTATT-3′; CD24, forward 5′-CTCCTACCCACGCAGA TTTATTC-3′ and 5′-AGAGTGAGACCACGAAGAGAC-3′; and hsa actin, forward 5′-ACA GAGCCTCGCCTTTGCC-3′ and reverse 5′-GAGGATGCCTCTCTTGCTCTG-3′. The 10-μl PCR reaction system consisted of 1 μl cDNA template, 5 μl Mix, 0.7 μl primer each, 0.05 μl ROX, and 2.55 μl RNase-free water. The reaction conditions were as follows: 95˚C for 2 min; 95˚C for 30 s for totally 40 cycles; and 60˚C for 30 s for totally 40 cycles.

## Western blot analysis

Cancer tissues and para-cancer tissues were subjected to RIPA lysis. After centrifugation, total proteins were obtained. The protein concentration was determined by the BCA method. Then the proteins were separated by SDS-PAGE and transferred onto the PVDF membrane (Cat# IPVH00010; Millipore). After blocking, the membrane was incubated with primary antibodies: anti-E-cadherin (1:10000 dilution; ab40772; Abcam), anti-β-catenin (1:5000 dilution; ab32572; Abcam), anti-vimentin (1:1000 dilution; ab8978; Abcam), anti-N-cadherin (1:5000 dilution; ab76011; Abcam), anti-CD44 (1:2000 dilution; ab157107; Abcam), anti-CD24 (1:200 dilution; MA5-11833; Invitrogen), and anti-β-actin (1:800 dilution; D110001; Shanghai Health Worker, Shanghai, China) at 4˚C overnight. After washing, the membranes were incubated with the secondary antibody at room temperature for 1 h. After washing, the signals were detected using Chemiluminescence Imaging System (ChemiScope Mini 3300; Shanghai Qinxiang Scientific Instrument Co., Ltd., Shanghai, China).

## Clinical pathological data collection

The subjected were divided into the group younger than 46 years and the group no younger than 46 years. According to the tumor sizes, the lesions were divided into the ≤1 cm, 2–5 cm, and ≥5 cm groups. According to the histological grades, the lesions were divided into the 1–2 grade and 3 grade groups. Moreover, the subjects were divided into the lymph node metastasis and none lymph node metastasis groups. According to the clinical stages, they were divided into the ≤2 stage group and the ≥3 stage group. Moreover, the subjects with the Luminal A and B types were classified as the hormone receptor-dependent group, while the subjects with the her-2 over-expression and triple-negative breast cancer were classified as the non-hormone receptor-dependent group.

## Statistical analysis

Data were expressed as mean±SD. The SPSS 21.0 software was used for statistical analysis. If the data conformed to the normal distribution, the single-factor analysis of variance was performed. Multiple comparisons were performed with the sidak method for the data with homogeneity of variance, while the Tamhane method was used for the data with the heterogeneity of variance. The data without normal distribution were first subjected to the logarithmical transform for normalization, and then the above-mentioned one-way analysis of variance method was performed for statistical analysis, or the Wilcoxon rank sum test was used. Correlation analysis of the expression index was performed with the Pearson correlation. $P < 0.05$ was considered as statistically significant.

## Results

### Analysis of clinicopathological characteristics of study subjects

The included 119 patients aged from 29 to 81 years, with a median age of 46 years. The pre-menopausal patients accounted for 60.5% (n = 72), and the postmenopausal patients accounted for 39.5% (n = 47). Moreover, 53.8% of patients had menarche later than 13 years old. All 119 specimens were confirmed by pathology, including 12 cases (10.1%) of *in situ* ductal carcinoma, 91 cases (76.5%) of invasive ductal carcinoma and invasive lobular carcinoma, and 16 cases (13.4%) of invasive ductal carcinoma with *in situ* carcinoma and invasive lobular carcinoma with *in situ* carcinoma. According to the histological grades, in these 119 specimens, there were 5 cases of grade 0, 10 cases of grade I, 71 cases of grade II, and 33 cases of grade III. All patients underwent the axillary lymphadenopathy, and 29 of them showed metastases. Moreover, there were 15 cases of Luminal A type, 42 cases of Luminal B type, 30 cases of her-2 over-expression type, and 32 cases of triple negative breast cancer.

### Expression characteristics of CSCs and EMT in these subjects

The representative sections of CEUS were marked before surgery and tissue specimens were collected from these marked sections. The expression levels of CSCs and EMT proteins in these specimens were investigated with the immunohistochemistry. Our results from the immunohistochemical analysis showed that the expression levels of CD44, β-catenin, and N-cadherin in the breast cancer tissues were up-regulated compared with the normal tissues. Moreover, the expression levels of CD24 and E-cadherin in the breast cancer tissues were down-regulated compared with the normal tissues. However, there were no significant differences in the expression levels of vimentin (Fig 2 and Table 1).

On the other hand, our results from the quantitative fluorescence analysis showed that, in the breast cancer tissues, the mRNA expression levels of *CD44*, *N-cadherin*, and *β-catenin* were significantly up-regulated, while the mRNA expression levels of *CD24* were significantly down-regulated, compared with the normal tissues adjacent to the cancer (Fig 3). There were no significant differences in the mRNA expression levels of *vimentin* and *E-cadherin* between the breast cancer tissues and adjacent normal tissues (Fig 3). Moreover, compared with normal tissues adjacent to cancer, the breast cancer tissues had significantly up-regulated protein expression levels of CD44, N-cadherin, and β-catenin, but significantly down-regulated protein expression levels of E-cadherin and CD24 (Fig 4A and 4B). No significant difference was observed in the Vimentin protein expression levels between the cancer and adjacent tissues.

### Correlation between CSC- and EMT-related markers and disease prognostic factors

Correlation between the relative expression levels of CSC- and EMT-related genes and the clinicopathological indicators of breast cancer was investigated. As shown in Table 2, our results showed that the CSCs expression was related to the histological grade, lymph node status, and hormone receptor status, with no relation with the age, tumor size, and clinical stage. Our results showed that the CSCs expression was elevated in the breast cancer tissues with higher histological grade, lymph node metastasis, and ER negative expression. Moreover, the EMT was related to the tumor size, histological grade, lymph node metastasis, clinical stage, and hormone receptor status, with no relation to the patient age. Furthermore, the breast cancer epithelial phenotype expression was up-regulated for the relatively small tumors, as well as the cases with no lymph node metastasis, lower clinical stage, and ER positive expression. However, the expression of interstitial phenotype tended to increase for

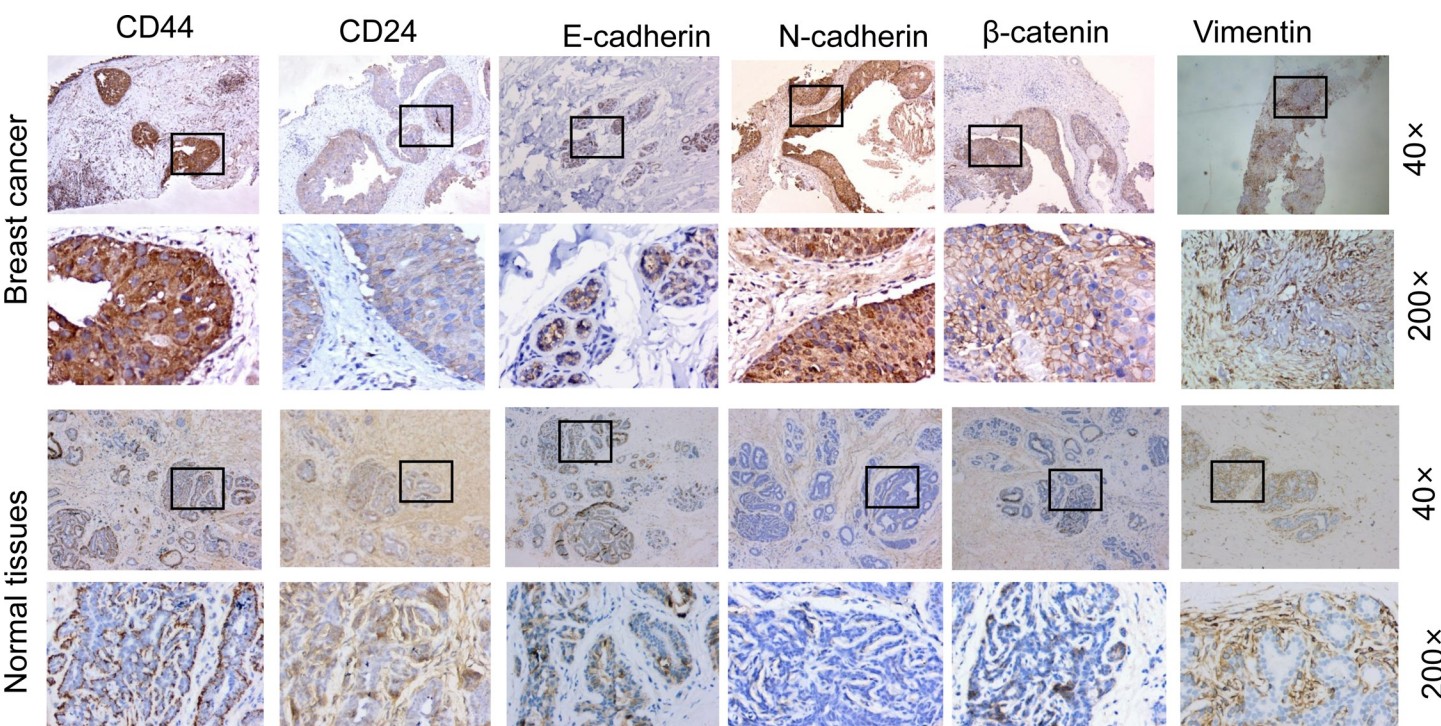

**Fig 2. Epithelial-mesenchymal transition (EMT) of breast cancer stem cells.** The expression levels EMT-related factors were detected with immunohistochemistry (Magnification 200× and 40×), including the CD44 (4 vs 2 points for the breast cancer and normal surrounding tissue groups, respectively), CD24 (2 vs 4points), E-cadherin (1 vs 3 points), N-cadherin (4 vs 0 points), β-catenin (3 vs 1 points), and Vimentin (3 vs 3 points).

the breast cancers with higher histological grade, lymph node metastasis, higher clinical stage, and ER negative expression.

## Correlation between tumor microcirculation perfusion characteristics and CSCs/EMT

The correlation between the CEUS performance of breast cancers and CSCs/EMT was investigated. The range of lesions on CEUS was beyond the range of two-dimensional ultrasound in 91 cases (76.5%) of breast cancer. Additionally, 37 cases (31.1%) of breast cancer showed perfusion defects. Using the surrounding normal tissue as control, the peak intensity, time to peak, starting time of enhancement, and the area under the curve were obtained. The time intensity curve of 90 cases (75.6%) showed fast-in mode and high enhancement (Fig 5). Our results showed that, the perfusion defect was observed after CEUS in the breast cancers, suggesting the morphological changes of ischemic necrosis in these breast cancer lesions, which was related to the expression of interstitial phenotype, while not related to the CEUS

**Table 1. Immunohistochemical score of CSCs and EMT in breast cancer tissues (n = 56).**

|  | CD24 | CD44 | E-cadherin | N-cadherin | β-catenin | Vimentin |
|---|---|---|---|---|---|---|
| **Breast cancer** | 1.333±1.047 | 2.931±1.241 | 1.250±0.500 | 2.017±1.235 | 2.586±1.200 | 3.431±0.797 |
| **Normal tissues** | 2.259±1.278 | 1.533±1.060 | 2.948±1.343 | 0.200±0.561 | 1.533±1.125 | 3.250±0.957 |
| **Z** | -2.531 | -3.828 | -4.681 | -4.634 | -2.984 | -1.284 |
| **P** | 0.011 | <0.001 | <0.001 | <0.001 | 0.003 | 0.199 |

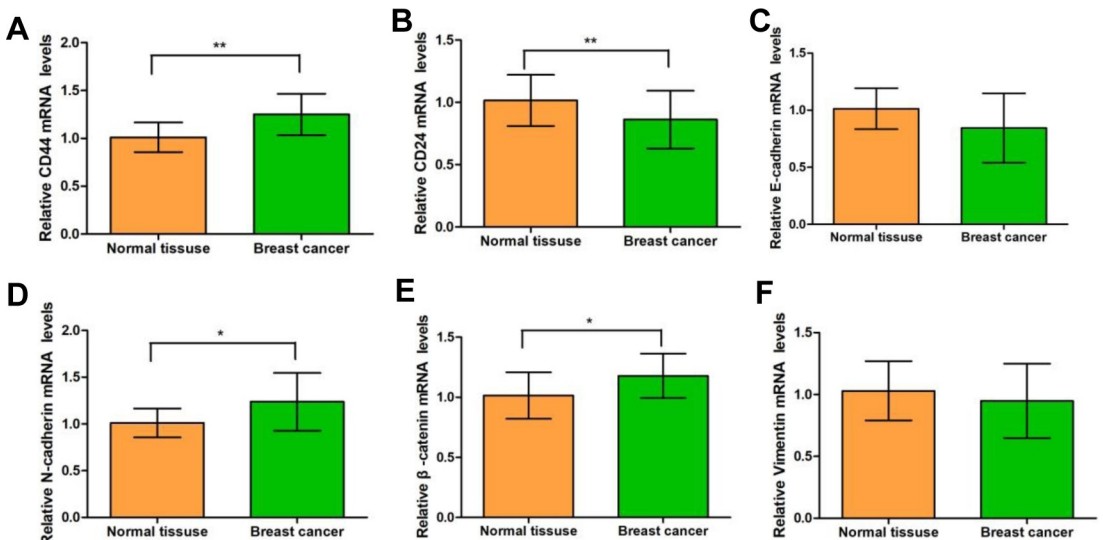

**Fig 3. The mRNA expression levels of CSC-related and EMT-related genes in breast cancer tissues (n = 63).** The mRNA expression levels of CD44 (A), CD24 (B), E-cadherin (C), N-cadherin (D), β-catenin (E), and Vimentin (F) were detected with the quantitative real-time PCR. Compared with the normal surrounding tissues, $^{*} P < 0.05$, $^{**} P < 0.01$.

quantitative parameters. Moreover, the breast cancer lesions with perfusion defect tended to have up-regulated expression of the interstitial marker N-cadherin, while there were no significant differences in the CSCs or the epithelial markers between these groups (Table 3). For the breast cancer with enlarged lesions after enhancement, the TIC tended to be fast-in and high-enhanced, with down-regulated expression of epithelial E-cadherin and up -regulated

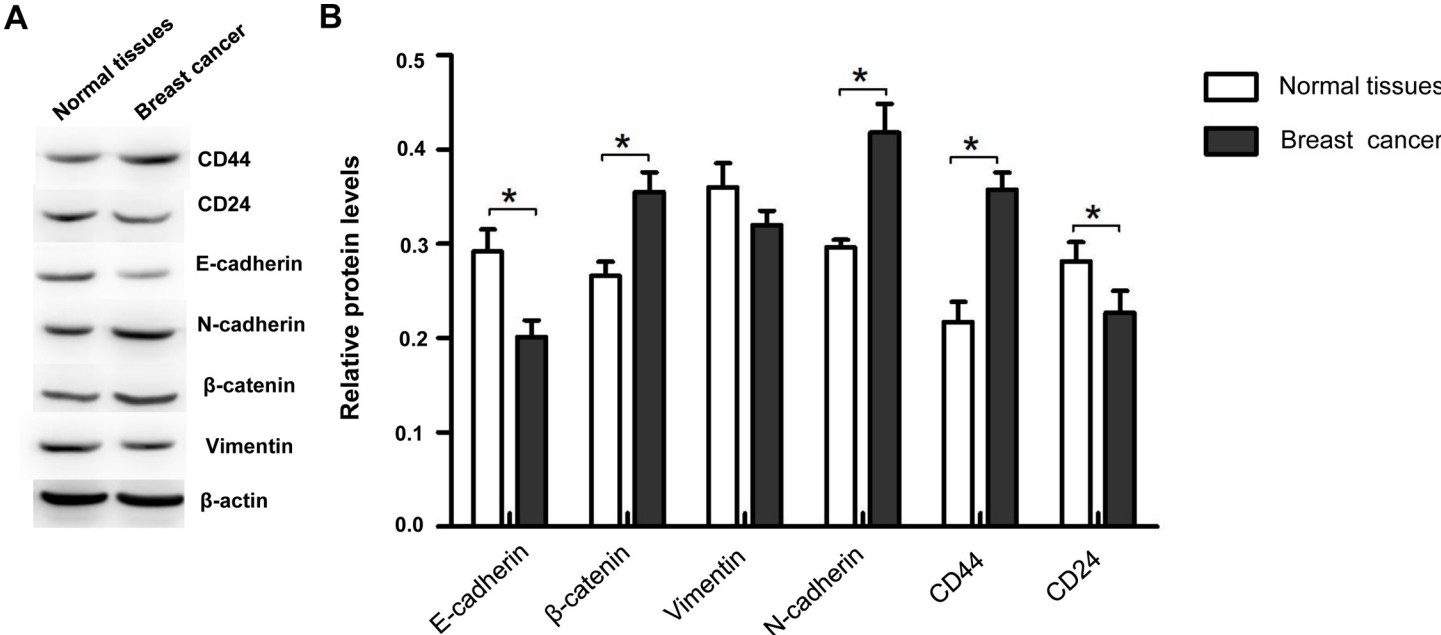

**Fig 4. The expression levels of CSCs-related and EMT-related proteins in breast cancer tissues (n = 9).** The protein expression levels of CD44, CD24, E-cadherin, N-cadherin, β-catenin, and Vimentin were detected with the Western blot analysis (A). Statistical analysis of the protein expression levels (B). Compared with the normal surrounding tissues, $^{*} P < 0.05$.

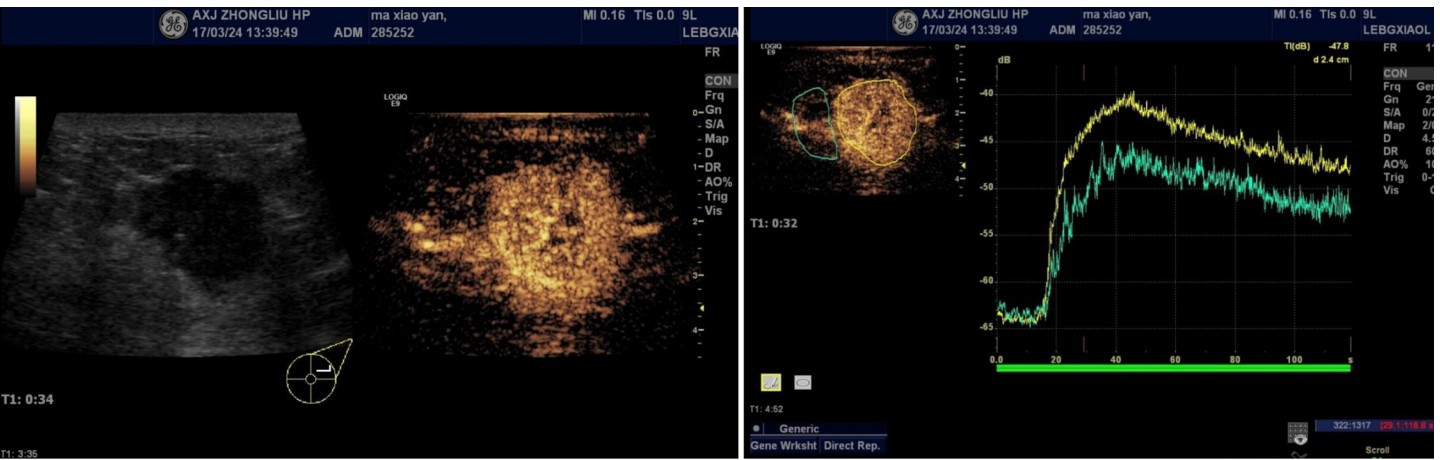

**Fig 5. Contrast-enhanced ultrasound (CEUS) of breast cancer.** The representative image of the CEUS in breast cancer was shown, with enlarged lesion range (expanding beyond that of the 2D ultrasound). Perfusion defect could be seen in the central area. Compared with the normal surrounding tissue, the TIC showed the fast-in pattern and high enhancement.

expression of interstitial N-cadherin (Table 4). Correlation analysis between the CEUS quantitative parameters of breast cancers and the relative expression levels of CSC- and EMT-related mRNAs showed that the peak intensity was positively related to the expression levels of the CSCs marker CD44 and the interstitial marker N-cadherin, while the starting time of enhancement was negatively related to the expression of interstitial marker N-cadherin. Moreover, the area under the curve was positively correlated with the expression levels of the CSCs marker CD44 and the interstitial marker N-cadherin, which was negatively correlated with the expression of the epithelial marker β-catenin. The time to peak was related to the expression levels of the interstitial markers Vimentin and N-cadherin, with no correlation with the epithelial marker E-cadherin or β-catenin (Table 5).

## Discussion

At present, the genetic tests of the penetrating tumor tissues have been commonly used in the disease diagnose and prognosis prediction. However, the heterogeneity of tumor structure often interferes with the accuracy of puncture [9], and CSCs often differentiate into cells with different heterogeneities [10]. The occurrence, development and metastasis of tumors all depend on the formation of microvessels, and the heterogeneity of tumors is also reflected in the microcirculation construction [11]. The 2D ultrasound combined with CEUS can reflect the microcirculation construction of breast cancer [12–14]. Breast cancer cells would cause severe damages to the original normal blood vessels and tissues, which would in turn induce the production of turbulent blood vessels [15]. The microvessel density of the lesions would be significantly increased, which significantly increases the blood flow to the cancer tissues, leading to a significant increase in the flow of the contrast agent in the cancer tissues in the early stage of perfusion [16]. The flow rate is significantly accelerated, the initial perfusion time and the peak time appear earlier, and the peak intensity is significantly increased [14]. However, the vascular wall in cancer tissue is often abnormally absent, resulting in a significant increase in the permeability of the blood vessel. A large number of tumor thrombi form in veins and lymphatic ducts, aggravating the degree of interstitial edema, and the tortuosity of blood vessels makes the contrast agent in the blood vessels. There is a large amount of retention in the vascular bed, so the clearance time of contrast agent in breast cancer is significantly extended,

**Table 2. Relationship between the relative mRNA expression levels of CSC- and EMT -related genes and clinicopathological indicators in breast cancer tissues.**

| | Age | | Statistical analysis | Lesion size | | | Statistical analysis | Histological grade | | | Statistical analysis | Lymph node metastasis | | Statistical analysis | Clinical stage | | Statistical analysis | Molecular pathology typing | | Statistical analysis |
|---|---|---|---|---|---|---|---|---|---|---|---|---|---|---|---|---|---|---|---|---|
| | <46 years | ≥46 years | | ≤1 cm | 2–5 cm | ≥5 cm | | Grade I | Grade II | Grade III | | Yes | No | | ≤2 stage | ≥3 stage | | Hormone receptor dependent | Non-hormone receptor dependent | |
| CD44 | 1.18 ±0.223 | 1.20 ±0.236 | T = -0.334, P = 0.739 | 1.213 ±0.113 | 1.267 ±0.242 | 1.221 ±0.179 | F = -0.312, P = 0.733 | 1.07 ±0.127 | 1.20 ±0.260 | 1.25 ±0.227 | F = -3.160, P = 0.049 | 1.200 ±0.234 | 1.015 ±0.215 | T = 3.473, P = 0.001 | 1.161 ±0.223 | 1.228 ±0.231 | T = -1.248, P = 0.216 | 1.081 ±0.210 | 1.281 ±0.267 | T = -2.202, P = 0.031 |
| E-cadherin | 0.87 ±0.247 | 0.89 ±0.316 | T = -0.236, P = 0.814 | 1.033 ±0.392 | 0.789 ±0.243 | 0.759 ±0.290 | F = -3.662, P = 0.033 | 0.87 ±0.318 | 0.85 ±0.199 | 0.95 ±0.342 | F = -0.745, P = 0.479 | 0.888 ±0.283 | 1.088 ±0.245 | T = 2.412, P = 0.019 | 0.827 ±0.246 | 0.942 ±0.310 | T = -1.741, P = 0.086 | 1.041 ±0.310 | 0.830 ±0.246 | T = -2.438, P = 0.018 |
| β-catenin | 1.13 ±0.214 | 1.15 ±0.179 | T = -0.265, P = 0.792 | 1.302 ±0.072 | 1.135 ±0.156 | 1.237 ±0.248 | F = -3.241, P = 0.047 | 1.12 ±0.176 | 1.09 ±0.218 | 1.19 ±0.175 | F = -1.900, P = 0.158 | 1.144 ±0.178 | 1.289 ±0.247 | T = 2.464, P = 0.016 | 1.200 ±0.212 | 1.088 ±0.164 | T = 2.464, P = 0.016 | 1.149 ±0.204 | 1.127 ±0.182 | T = -0.464, P = 0.644 |
| Vimentin | 0.99 ±0.314 | 0.94 ±0.264 | T = -0.705, P = 0.483 | 1.137 ±0.469 | 0.920 ±0.267 | 0.936 ±0.290 | F = -1.374, P = 0.262 | 1.00 ±0.157 | 1.00 ±0.371 | 0.91 ±0.253 | F = -0.691, P = 0.505 | 0.958 ±0.295 | 0.979 ±0.282 | T = 2.249, P = 0.804 | 0.965 ±0.310 | 0.962 ±0.268 | T = 0.034, P = 0.973 | 0.987 ±0.331 | 0.926 ±0.202 | T = -0.860, P = 0.393 |
| N-cadherin | 1.13 ±0.179 | 1.23 ±0.365 | T = -1.389, P = 0.169 | 1.419 ±0.635 | 1.187 ±0.219 | 1.286 ±0.298 | F = -1.713, P = 0.191 | 0.97 ±0.123 | 1.09 ±0.125 | 1.41 ±0.361 | F = -18.647, P < 0.001 | 1.201 ±0.319 | 1.004 ±0.147 | T = 2.323, P = 0.023 | 1.105 ±0.172 | 1.260 ±0.366 | T = -2.323, P = 0.023 | 1.005 ±0.171 | 1.260 ±0.366 | T = -2.260, P = 0.029 |
| CD24 | 0.88 ±0.159 | 0.92 ±.280 | T = -0.744, P = 0.460 | 0.776 ±0.125 | 0.836 ±0.236 | 0.967 ±0.236 | F = -2.035, P = 0.141 | 0.86 ±0.174 | 0.89 ±0.207 | 0.93 ±0.272 | F = -0.454, P = 0.637 | 0.881 ±0.228 | 0.984 ±0.220 | T = 1.569, P = 0.121 | 0.906 ±0.256 | 0.898 ±0.196 | T = 0.135, P = 0.893 | 0.905 ±0.177 | 0.897 ±0.296 | T = -0.152, P = 0.880 |

**Table 3.  Correlation between perfusion defect after breast cancer imaging and relative expression of CSC- and EMT-related genes.**

| | E-cadherin | β-catenin | Vimentin | N-cadherin | CD44 | CD24 | Peak intensity | Starting time of enhancement | Area under the curve | Time to peak |
|---|---|---|---|---|---|---|---|---|---|---|
| **No perfusion defect** | 0.780 ±0.223 | 1.151 ±0.197 | 0.974 ±0.366 | 1.159 ±0.199 | 1.264 ±0.248 | 0.822 ±0.160 | 20.670 ±7.968 | 15.833±4.594 | 1122.021 ±552.451 | 23.633 ±8.177 |
| **Perfusion defect** | 0.925 ±0.371 | 1.212 ±0.165 | 0.915 ±0.191 | 1.334 ±0.388 | 1.233 ±0.171 | 0.910 ±0.296 | 24.692 ±7.827 | 13.333±4.733 | 1275.142 ±533.454 | 20.958 ±6.617 |
| **T/Z value** | -1.688 | -1.204 | -0.122 | -2.142 | 0.506 | -1.036 | -1.858 | 1.960 | -1.028 | 1.298 |
| ***P*** | 0.100 | 0.234 | 0.903 | 0.037 | 0.615 | 0.300 | 0.069 | 0.055 | 0.309 | 0.200 |

and the area under the curve is significantly increased [14]. In our previous study, many qualitative indicators and quantitative parameters of CEUS had been investigated, which had been shown to reflect the tumor heterogeneity of breast cancer microcirculation construction, therefore screening out the most valuable prognostic factors for breast cancer [4]. In this study, according to the macroscopic images, the representative cancer tissues were screened out based on the CEUS indicators, and subjected to the molecular biological test, and the correlation between the imaging indicators with the breast cancer CSCs and EMT indicators was analyzed, further evaluating the ability of CEUS for the assessment of breast cancer prognosis.

It is indicated that CSCs and the EMT may be linked, and their key signal transduction pathways are intersecting, both playing important roles in tumor metastasis and recurrence [17]. Moreover, the EMT is closely also related to the function of CSCs [17]. In this study, our results showed that, β-catenin, as an epithelial phenotype, had a higher expression rate in the study cases, presumably related to the higher proportion of triple-negative breast cancer in these cases. It has been shown that the β-catenin would be abnormally increased in the tumor invasion and metastasis, as well as over-expressed in the triple negative breast cancers [18]. Among the prognostic factors, the breast cancers with large tumor diameter, high histological grade, cerebral fossa lymph node metastasis, high clinical stage, and negativeness for hormone receptor tended to have up-regulated EMT expression. Moreover, the breast cancers with high histological grade, cerebral fossa lymph node metastasis, and negative hormone receptors expression tended to have up-regulated expression of CSCs. These results suggest that, the up-regulated CSC and EMT expressions might be associated with the rapid proliferation of cancer cells, enlarged tumor lesions, lymphatic and hematogenous metastases, and poor prognosis. It has been shown that the increased expressions of CD44 in the breast cancer tissues determine the tumor progression and metastasis, and the CD44 could weaken the adhesion between tumor cells and enhance the matrix adhesion, promoting the tumor cell migration and metastasis [19]. The expression of E-cadherin would be inhibited by certain factors, leading to the destruction of the interepithelial connection and the weakening of the adhesion between

**Table 4.  Relationship between the enhancement and expansion of breast cancer after angiography and the relative mRNA expression of CSC- and EMT-related genes.**

| | E-cadherin | β-catenin | Vimentin | N-cadherin | CD44 | CD24 | Peak intensity | Starting time of enhancement | Area under the curve | Time to peak |
|---|---|---|---|---|---|---|---|---|---|---|
| **No enhancement enlargment** | 0.923 ±0.325 | 1.198 ±0.193 | 1.022 ±0.389 | 1.055 ±0.198 | 1.265 ±0.196 | 0.783 ±0.108 | 19.357 ±6.955 | 16.739±4.191 | 1027.495 ±556.650 | 25.913 ±8.597 |
| **Enhancement enlargment** | 0.738 ±0.240 | 1.164 ±0.180 | 0.893 ±0.202 | 1.297 ±0.360 | 1.239 ±0.232 | 0.919 ±0.280 | 24.759 ±8.198 | 13.226±4.695 | 1310.699 ±510.972 | 19.871 ±5.578 |
| **T/Z value** | 2.306 | 0.666 | -0.875 | -2.424 | 0.436 | -1.854 | -2.550 | 2.844 | -1.939 | 3.129 |
| ***P*** | 0.025 | 0.508 | 0.382 | 0.019 | 0.665 | 0.064 | 0.014 | 0.006 | 0.058 | 0.003 |

**Table 5. Correlation analysis of CEUS parameters of breast cancer with relative mRNA expressions of CSC- and EMT-related genes.**

| | Correlation coefficient | | | | | |
|---|---|---|---|---|---|---|
| | E-cadherin | β-catenin | Vimentin | N-cadherin | CD44 | CD24 |
| Peak intensity | 0.023 | 0.256 | -0.091 | 0.384 | 0.445 | 0.032 |
| *P* | 0.826 | 0.014 | 0.389 | <0.001 | <0.001 | 0.766 |
| Starting time of enhancement | -0.090 | -0.024 | 0.170 | -0.288 | -0.164 | -0.093 |
| *P* | 0.395 | 0.823 | 0.106 | 0.005 | 0.117 | 0.375 |
| Area under the curve | -0.132 | -0.402 | 0.066 | 0.500 | 0.496 | -0.044 |
| *P* | 0.208 | <0.001 | 0.530 | <0.001 | <0.001 | 0.680 |
| Time to peak | -0.042 | -0.009 | -0.232 | -0.246 | -0.094 | -0.161 |
| *P* | 0.689 | 0.936 | 0.026 | 0.018 | 0.373 | 0.126 |

cancer cells. The expression of N-cadherin would lead to the interstitial characteristics, which makes it easy to escape from the primary site and metastasize. Tumor cells would acquire the stem cell-like characteristics during the EMT process, which also proves the close relationship between these two processes [20]. These results suggest that the expression of CSCs and EMT are closely related to the disease prognosis.

The proliferation, differentiation, and metastasis of breast cancers are associated with the tumor blood vessels, and the qualitative and quantitative parameters of CEUS are mainly the macroscopic manifestations of tumor blood vessels [4]. In this study, the correlation between CEUS performance of breast cancer lesions and CSC/EMT was investigated, thus suggesting the relationship between tumor microcirculation perfusion characteristics and CSC/EMT. Perfusion defect in breast cancers after CEUS is also a manifestation of structural heterogeneity. The structural and functional defects of blood vessels in tumor tissues would cause blood flow disturbances and uneven perfusion pattern in the tumor lesions. The rapid proliferation of tumor cells would lead to increased demand for oxygen, finally resulting in hypoxia [20]. Hypoxia can induce tumor cells to develop EMT, and at the same time acquire the characteristics of CSCs, synergistically contributing to the tumor invasion and metastasis [21]. Hypoxia often occurs in larger breast cancers with necrosis, which would be manifested as perfusion defect in the CEUS features. In this study, our results confirmed the high expression levels of interstitial phenotype N-cadherin in the breast cancer with perfusion defect. It has also been confirmed in a previous study that the perfusion defects are related to older age, higher histological grade, and later clinical stage [5], suggesting that it might be related to poor disease prognosis. In the solid tumors, the CSC production, their ability to self-renew, and the undifferentiated state maintenance require the activation of tumor cells under hypoxic conditions to acquire new functional properties [21]. Hypoxia plays an important role in the phenotype and function of CSCs. However, in this study, our results showed that the perfusion defect was not related to other CEUS parameters, suggesting that there may be no difference in the incidence of hypoxic necrosis between breast cancer with rich blood supply and with poor blood supply, that is, breast cancer with perfusion defect is not necessarily the breast cancer with deficient blood supply.

On the contrary, it has been shown that the enhancement and expansion after CEUS often occur in breast cancers with high perfusion and blood supply. The part of the enhancement beyond the lesion area of 2D ultrasound is the marginal zone of breast cancer, i.e., the proliferation and infiltration area. In these proliferation and infiltration areas, the structure is abnormally arranged, and the tumors intersperses into the normal tissues, often accompanied by the hyperplastic fibrous tissue, lymphatic vessels, and blood vessels, resulting in unclear mass

boundary [18]. Due to the large number of tortuous and irregularly nourishing blood vessels, vigorously dividing vascular endothelial cells, and arteriovenous fistulas, the CEUS enhancement shows that this area is the most aggressive area in the breast cancer [22]. In this study, our results showed that the breast cancer with enlarged lesions after enhancement was more likely to undergo the EMT. Because EMT is usually associated with poor prognosis, the results of this study also indirectly indicate that enlarged lesions after enhancement are related to poor prognosis of breast cancer.

It has been shown that the TICs for the breast cancer characterized by the fast-in pattern, high enhancement, and high perfusion, are characterized by the high expressions of CSCs and EMT. Moreover, the correlation between the TIC and the interstitial phenotype is stronger than with the epithelial phenotype. When CSCs and EMT are highly expressed, the breast tumors would proliferate rapidly and aggressively. The occurrence of EMT in CSCs will promote the proliferation and migration of fibroblasts, thus inducing the angiogenesis, abundant tumor blood supply, fast blood flow, and elevated average blood volume, with thin tumor vessel wall. Moreover, the arteriovenous fistula is easy to form, which would cause the abnormal high-speed blood flow, as indicated by the fast-in pattern, high enhancement and high perfusion in the TIC.

## Conclusions

In conclusion, our results showed that the CEUS performance of breast cancers had a certain correlation with CSCs and EMT. Some of the CEUS indicators may contribute to the prognosis assessment of breast cancer. The enlarged lesions after enhancement, perfusion defect, fast-in pattern, and high enhancement might indirectly indicate the poor prognosis of breast cancer. However, further in-depth studies concerning the application of CEUS imaging omics in clinic, with enlarged sample size and standardized operating procedures, are still needed in the future.

## Supporting information

**S1 Checklist. STROBE statement—checklist of items that should be included in reports of observational studies.**
(DOCX)

## Author Contributions

**Conceptualization:** Jianbing Ding, Fucheng Ma.

**Data curation:** Xiaoling Leng, Guofu Huang, Siyi Li, Miaomiao Yao.

**Formal analysis:** Xiaoling Leng, Guofu Huang, Siyi Li, Miaomiao Yao.

**Funding acquisition:** Jianbing Ding.

**Investigation:** Xiaoling Leng, Guofu Huang, Siyi Li, Miaomiao Yao.

**Methodology:** Xiaoling Leng, Guofu Huang, Siyi Li, Miaomiao Yao.

**Software:** Xiaoling Leng.

**Supervision:** Jianbing Ding, Fucheng Ma.

**Validation:** Jianbing Ding, Fucheng Ma.

**Writing – original draft:** Xiaoling Leng.

**Writing – review & editing:** Jianbing Ding, Fucheng Ma.

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
