## [Decision Letter · Decision Letter 0]

16 Nov 2020

PONE-D-20-18825

Correlation of breast cancer microcirculation construction with tumor stem cells (CSCs) and epithelial-mesenchymal transition (EMT) based on contrast-enhanced ultrasound (CEUS)

PLOS ONE

Dear Dr. Ding,

Thank you for submitting your manuscript to PLOS ONE. After careful consideration, we feel that it has merit but does not fully meet PLOS ONE’s publication criteria as it currently stands. Therefore, we invite you to submit a revised version of the manuscript that addresses the points raised during the review process.

Please see the reviewer reports below. The reviewers have requested further information on the aims and motivations, as well as further details in the methodology to help ensure that the study is reproducible by another researcher (PLOS ONE publication criterion #3). We also ask that you ensure that your manuscript is copyedited for English grammar and use.

We note that one reviewer has suggested that some citations may not be appropriate, however, there is no journal requirement to remove any citations that helped inform the study.

We look forward to receiving your revised manuscript.

Kind regards,

Hanna Landenmark

Associate Editor

PLOS ONE

Journal Requirements:

2. In the Methods section, please provide the source, product number and any lot numbers of the primary antibodies used for the immunohistochemical analysis conducted for your study.

3. In your Methods section, please provide additional information about the participant recruitment method and the demographic details of your participants. Please ensure you have provided sufficient details to replicate the analyses such as:  a) a table of relevant demographic details, b) a statement as to whether your sample can be considered representative of a larger population, c) a description of how participants were recruited, and d) descriptions of where participants were recruited and where the research took place.

4. Please provide a sample size and power calculation in the Methods, or discuss the reasons for not performing one before study initiation.

5. At this time, we ask that you please provide scale bars on the microscopy images presented in Figure 1 and refer to the scale bar in the corresponding Figure legend.

6.Thank you for stating the following in the Funding Section of your manuscript:

[This work was supported by the Natural Science Foundation of China (No.

411 81660496). The funders had no role in study design, data collection and analysis,

412 decision to publish, or preparation of the manuscript.]

 [The funders had no role in study design, data collection and analysis, decision to publish, or preparation of the manuscript.]

7.We note that you have indicated that data from this study are available upon request. PLOS only allows data to be available upon request if there are legal or ethical restrictions on sharing data publicly. For more information on unacceptable data access restrictions, please see http://journals.plos.org/plosone/s/data-availability#loc-unacceptable-data-access-restrictions.

Reviewers' comments:

Reviewer's Responses to Questions

**Comments to the Author**

1. Is the manuscript technically sound, and do the data support the conclusions?

Reviewer #1: Partly

Reviewer #2: Partly

2. Has the statistical analysis been performed appropriately and rigorously? 

Reviewer #1: No

Reviewer #2: I Don't Know

3. Have the authors made all data underlying the findings in their manuscript fully available?

Reviewer #1: Yes

Reviewer #2: Yes

4. Is the manuscript presented in an intelligible fashion and written in standard English?

Reviewer #1: No

Reviewer #2: No

5. Review Comments to the Author

Reviewer #1: The authors have demonstrated that the correlation between the parameters of less invasive CEUS and the expression level of CSC/EMT markers in breast cancer by IHC and RT-PCR. However, the experimental results are not sufficient to state their conclusions. The authors must evaluate IHC either in the same cell (double staining) or by the staining using serial sections. RT-PCR should use RNA extracted from cancer cells or normal cells recovered by laser microdissection, rather than RNA extracted from whole tissues. Furthermore, it is necessary to use the TaqMan probe method, which has a high quantitative yield, instead of the cybr green method, which is a semi-quantitative method.

Minor points:

1) Please insert scale bars in Fig.1.

2)The information of antibodies of IHC should be written down in Materials and Methods.

3)Please describe the detailed statistical analysis methods in Materials and Methods.

4)There are inappropriate citations, such as quoting a lung cancer paper even though it is a breast cancer citation. Please review the selection of cited papers.

5) There are many misspellings. Please check the spelling.

Reviewer #2: Contrast-enhanced ultrasound (CEUS) procedure carried out with new medical equipment and new-generation micro-bubble contrast reagents has been suggested to increase the accuracy in providing information on the microcirculation. The overall aim of the research described in this manuscript is to assess whether CEUS-based features may provide prognostic type of information. In particular, CEUS-based microcirculation features are examined for correlation with cancer-stem cell (CSC)-like subpopulations including those undergoing epithelial-mesenchymal transition (EMT). The rationale for focusing on tumor cells with CSC and EMT-like characteristics relates to potential roles in tumor heterogeneity and aggressiveness. The manuscript including the background, methods, and results would benefit from better language use and inclusion of information in terms of providing a clearer understanding of the rationale and the objectives of the study, the methodologies used, data interpretation including implications.

Specific Comments-

1. Some of the information presented in the discussion should be considered to be moved to the background to help provide a basis for the study.

2. Several published studies (eg Ji et al, 2017. Clin Hemorheol Microcirc, Zhao et al, 2017 Onco Targets Ther, Cao et al, 2014. Ultrasound Med Biol, Wang et al, 2017, PLoS One, and Vraka et al, 2918 In Vivo) have assessed the correlation between CEUS-based parameters and prognostic markers in breast cancer should be alluded to and referenced in the background and/or discussion.

3. The reason for use of 2D US prior to the CEUS should be included in the method.

4. A figure that depicts the various parameters obtained related to the CEUS describe on page 6 in the methods should be included (see ). The functional implications of these parameters in terms of circulation/perfusion should be clearly stated to help understand their significance when described in the result sections. Use of full names (and not abbreviations) is recommended throughout the manuscript.

5. A rationale should be provided for the criteria used to follow the various biomarkers described in the IHC section of the method. The nature of the antibodies used what they recognize on target proteins should be included. What is the rationale for looking for cytoplasmic localization of CD24? β-catenin localization in cytoplasmic/nucleus can relate to WNT-pathway activation and potentially EMT, given this was this aspect inspected? How epithelial/carcinoma vs stromal cells distinguished? Where antibodies for cytokeratins used?

6. EMT induction requires the expression of transcription factors. Thus, to better show the relationship of the biomarkers used, it will be important to assess the expression (correlation) of some of EMT-TF with the biomarkers and the CEUS-features tested in the current study.

6. PLOS authors have the option to publish the peer review history of their article (what does this mean?). If published, this will include your full peer review and any attached files.

Reviewer #1: No

Reviewer #2: No

---

## [Author Response · Author response to Decision Letter 0]

11 May 2021

Dear editor,

On behalf of all co-authors, I would like to thank you for your letter concerning our manuscript. We greatly appreciate you and the reviewers for the critical reading of our manuscript and giving us the favorable comments and instructive suggestions.

We have carefully proof-read and revised the manuscript according to the reviewers’ comments and the editorial notes. Here we submit the revised manuscript. In the following pages, the responses to the comments are described point-by-point. 

Once again, we want to extend our appreciation to you and the reviewers for the valuable and helpful comments. We sincerely hope that the manuscript has been revised to your satisfaction, and we would be grateful if the manuscript could be considered for publication.

In addition, we would like to update our Funding Statement as follows:

This work was supported by Natural Science Foundation of Xinjiang Uygur Autonomous Region (2018D01C276).

Sincerely yours,

Jianbing Ding

Basic Medical College, Xinjiang Medical University, No. 393 Xinyi Road, Urumqi 830011, Xinjiang, China. Tel: 86-18999230065. Email: dingjb1234@ aliyun.com & 1601379937@qq.com

The reviewer’s comments and the authors’ responses:

Reviewer #1: The authors have demonstrated that the correlation between the parameters of less invasive CEUS and the expression level of CSC/EMT markers in breast cancer by IHC and RT-PCR. However, the experimental results are not sufficient to state their conclusions. 

Response: To validate our conclusions, we have further performed Western blot analysis of CSC-related and EMT-related markers in tumor and tumor adjacent tissues. The results showed that compared with normal tissues adjacent to cancer, the breast cancer tissues had significantly up-regulated protein expression levels of CD44, N-cadherin, and β-catenin, but significantly down-regulated protein expression levels of E-cadherin and CD24 (Fig. 3A and 3B). No significant difference was observed in the Vimentin protein expression levels between the cancer and adjacent tissues. We have added this result to the revised Fig. 3 and modified the revised manuscript accordingly.

However, further studies are still needed to validate our results. 

The authors must evaluate IHC either in the same cell (double staining) or by the staining using serial sections. 

Response: In immunohistochemistry, the staining color position and depth indicated the expression levels and patterns, and it was difficult to perform double staining on the very one section. Therefore, these indicators were subjected to IHC staining on serial sections.

However, the reviewer raised a very good point. We will do as suggested in future studies.

RT-PCR should use RNA extracted from cancer cells or normal cells recovered by laser microdissection, rather than RNA extracted from whole tissues. 

Response: The tumor samples were obtained under the guidance of B-ultrasound. The hyperplastic gland tissues were used as control. Due to limited funding, we did not recover the cells by laser microdissection. However, the reviewer raised a very good point. We will do as suggested in future studies.

Furthermore, it is necessary to use the TaqMan probe method, which has a high quantitative yield, instead of the cybr green method, which is a semi-quantitative method.

Response: The quantitative real-time PCR was performed to detect the mRNA expression levels of the target genes. However, the reviewer raised a very good point. We will do as suggested in future studies.

Minor points:

1) Please insert scale bars in Fig.1.

Response: In fact, the figures were taken at the magnification of 200×, which has been added in the figure legends.

2) The information of antibodies of IHC should be written down in Materials and Methods.

Response: According to the comment, we have added the information of IHC antibodies in Materials and Methods section of the revised manuscript. 

3) Please describe the detailed statistical analysis methods in Materials and Methods.

Response: If the data conformed to the normal distribution, the single-factor analysis of variance was performed. Multiple comparisons were performed with the sidak method for the data with homogeneity of variance, while the Tamhane method was used for the data with the heterogeneity of variance. The data without normal distribution were first subjected to the logarithmical transform for normalization, and then the above-mentioned one-way analysis of variance method was performed for statistical analysis, or the Wilcoxon rank sum test was used. Correlation analysis of the expression index was performed with the Pearson correlation. 

4) There are inappropriate citations, such as quoting a lung cancer paper even though it is a breast cancer citation. Please review the selection of cited papers.

Response: Sorry for the inappropriate citations. According to the comment, corresponding changes have been made herein.

5) There are many misspellings. Please check the spelling.

Response: According to the comment, we have further gotten editorial help to improve the English writing of the revised manuscript. Please check!

Reviewer #2: Contrast-enhanced ultrasound (CEUS) procedure carried out with new medical equipment and new-generation micro-bubble contrast reagents has been suggested to increase the accuracy in providing information on the microcirculation. The overall aim of the research described in this manuscript is to assess whether CEUS-based features may provide prognostic type of information. In particular, CEUS-based microcirculation features are examined for correlation with cancer-stem cell (CSC)-like subpopulations including those undergoing epithelial-mesenchymal transition (EMT). The rationale for focusing on tumor cells with CSC and EMT-like characteristics relates to potential roles in tumor heterogeneity and aggressiveness. The manuscript including the background, methods, and results would benefit from better language use and inclusion of information in terms of providing a clearer understanding of the rationale and the objectives of the study, the methodologies used, data interpretation including implications.

Specific Comments-

1. Some of the information presented in the discussion should be considered to be moved to the background to help provide a basis for the study.

Response: According to the comment, corresponding changes have been made herein.

2. Several published studies (eg Ji et al, 2017. Clin Hemorheol Microcirc, Zhao et al, 2017 Onco Targets Ther, Cao et al, 2014. Ultrasound Med Biol, Wang et al, 2017, PLoS One, and Vraka et al, 2918 In Vivo) have assessed the correlation between CEUS-based parameters and prognostic markers in breast cancer should be alluded to and referenced in the background and/or discussion.

Response: According to the comment, we have added the following reference to the Discussion section. Please check!

Reference:

Vraka I, Panourgias E, Sifakis E, Koureas A, Galanis P, Dellaportas D, Gouliamos A, Antoniou A. In Vivo.Correlation Between Contrast-enhanced Ultrasound Characteristics (Qualitative and Quantitative) and Pathological Prognostic Factors in Breast Cancer. 2018;32(4):945-954. Epub 2018/07/30. doi: 10.21873/invivo.11333. PubMed PMID: 29936484;PubMed Central PMCID: PMC6117754. 

3. The reason for use of 2D US prior to the CEUS should be included in the method.

Response: In fact, for the CEUS, the contrast mode should be conducted on the basis of a clear two-dimensional ultrasound image. Although the results needed not to be analyzed for the two-dimensional ultrasound, the operation method would be necessary, which represented the diagnostic rule for contrast ultrasound. However, these methodological information needed not to be explained in the study. 

4. A figure that depicts the various parameters obtained related to the CEUS describe on page 6 in the methods should be included (see ). The functional implications of these parameters in terms of circulation/perfusion should be clearly stated to help understand their significance when described in the result sections. Use of full names (and not abbreviations) is recommended throughout the manuscript.

Response: According to the comment, corresponding information has been added herein. For example, we have used the full names instead of abbreviations according to the suggestion.

5. A rationale should be provided for the criteria used to follow the various biomarkers described in the IHC section of the method. The nature of the antibodies used what they recognize on target proteins should be included. What is the rationale for looking for cytoplasmic localization of CD24? β-catenin localization in cytoplasmic/nucleus can relate to WNT-pathway activation and potentially EMT, given this was this aspect inspected? How epithelial/carcinoma vs stromal cells distinguished? Where antibodies for cytokeratins used?

Response: According to the comment, corresponding information concerning the primary antibodies and the criteria for expression evaluation have been added in the revised manuscript. 

Moreover, based on the Human-Protein-Atlas database, CD24 is located in the cytoplasm. The activation of the WNT pathway would cause the translocation of β-catenin from the cytoplasm to the nucleus [1]. In the nucleus, β-catenin combines with the LEF/TCF transcription factor to replace the auxiliary inhibitor, and this combination recruits the additional auxiliary activators of the Wnt target genes to promote the process of EMT [2]. There have been articles [3] reporting that CD24 is the target of Wnt in colorectal cancers. Therefore, from database predictions and previous literature, the WNT pathway activation and potential EMT would affect the CD24 cytoplasmic localization.

In addition, through information retrieving, the keratin (epithelial cells) and vimentin (stromal cells) are used to distinguish the epithelial cells from the stromal cells.

References:

[1] Borgal L, Habbig S, Hatzold J, Liebau MC, Dafinger C, Sacarea I, Hammerschmidt M, Benzing T, Schermer B. The ciliary protein nephrocystin-4 translocates the canonical Wnt regulator Jade-1 to the nucleus to negatively regulate β-catenin signaling. J Biol Chem. 2012 Jul 20;287(30):25370-80.

[2] Kobayashi W, Ozawa M. The epithelial-mesenchymal transition induced by transcription factor LEF-1 is independent of β-catenin. Biochem Biophys Rep. 2018 Jun 12;15:13-18. 

[3] Chen YC, Lee TH, Tzeng SL. Reduced DAXX Expression Is Associated with Reduced CD24 Expression in Colorectal Cancer. Cells. 2019 Oct 12;8(10):1242.

6. EMT induction requires the expression of transcription factors. Thus, to better show the relationship of the biomarkers used, it will be important to assess the expression (correlation) of some of EMT-TF with the biomarkers and the CEUS-features tested in the current study.

Response: Based on the PROMO database, Genecards database and UCSC database, the transcription factors related to EMT indicators were predicted. The common transcription factors included the SP1, YY-1, etc. Through literature searching, the transcription factors were all related to EMT, which regulated the process of EMT through modulating the gene transcription. It has been reported that the prepared nanobubbles showed high affinity and specificity to breast cancer cells and tumors with the neuropeptide YY 1 receptors, with minimal toxicity and damages to the organs [1]. Biodegradable photoluminescent nanobubbles were used as ultrasound contrast agents for targeted breast cancer imaging. The transcription factor prediction results were obtained as attached files.

Reference: 

[1] Li J, Tian Y, Shan D, Gong A, Zeng L, Ren W, Xiang L, Gerhard E, Zhao J, Yang J, Wu A. Neuropeptide Y Y1 receptor-mediated biodegradable photoluminescent nanobubbles as ultrasound contrast agents for targeted breast cancer imaging. Biomaterials. 2017 Feb;116:106-117.

---

## [Decision Letter · Decision Letter 1]

12 Jul 2021

PONE-D-20-18825R1

Correlation of breast cancer microcirculation construction with tumor stem cells (CSCs) and epithelial-mesenchymal transition (EMT) based on contrast-enhanced ultrasound (CEUS)

PLOS ONE

Dear Dr. Ding,

Thank you for submitting your manuscript to PLOS ONE. After careful consideration, we feel that it has merit but does not fully meet PLOS ONE’s publication criteria as it currently stands. Therefore, we invite you to submit a revised version of the manuscript that addresses the points raised during the review process.

As evident from the Reviewer's comments, it is critical to re-address the images in Figure 1. It is very hard to see how the images provided come from serial sections, or how they correlate with each other, so that one can extract meaningful conclusions. Please provide a revised Figure 1 with appropriate images, and/or low magnifications of the images to allow for evaluation of location of panels and potential co-localizations, as well as the assessment of the same tumor.

We look forward to receiving your revised manuscript.

Kind regards,

Stella E. Tsirka

Academic Editor

PLOS ONE

Additional Editor Comments (if provided):

As evident from the Reviewer's comments, it is critical to re-address the images in Figure 1. It is very hard to see how the images provided come from serial sections, or how they correlate with each other, so that one can extract meaningful conclusions. Please provide a revised Figure 1 with appropriate images, and/or low magnifications of the images to

Reviewers' comments:

Reviewer's Responses to Questions

**Comments to the Author**

1. If the authors have adequately addressed your comments raised in a previous round of review and you feel that this manuscript is now acceptable for publication, you may indicate that here to bypass the “Comments to the Author” section, enter your conflict of interest statement in the “Confidential to Editor” section, and submit your "Accept" recommendation.

Reviewer #1: All comments have been addressed

2. Is the manuscript technically sound, and do the data support the conclusions?

Reviewer #1: No

3. Has the statistical analysis been performed appropriately and rigorously? 

Reviewer #1: No

4. Have the authors made all data underlying the findings in their manuscript fully available?

Reviewer #1: Yes

5. Is the manuscript presented in an intelligible fashion and written in standard English?

Reviewer #1: No

6. Review Comments to the Author

Reviewer #1: They have not essentially revised for my major concerns.

My biggest concerns are below. Although authors described that they use the serial section for IHC, but as far as I can see in the picture in Figure 1, I can not think that they have observed the same tumor or normal site in the continuous section. The photos should be shown the same tumor or normal site in the serial section.

In figure4, the amounts of N-cadherin, Beta-catenin, and Vimentin proteins in these representative photos are not consistent with these quantitative graphs.

Thus, I think that their experimental results are inadequate to draw their conclusions.

7. PLOS authors have the option to publish the peer review history of their article (what does this mean?). If published, this will include your full peer review and any attached files.

Reviewer #1: No

---

## [Author Response · Author response to Decision Letter 1]

25 Aug 2021

Dear editor,

 We are resubmitting the Manuscript ID PONE-D-20-18825R1 entitled Correlation of breast cancer microcirculation construction with tumor stem cells (CSCs) and epithelial-mesenchymal transition (EMT) based on contrast-enhanced ultrasound (CEUS) to PLOS ONE. Our responses to the comments by the reviewers are outlined below. Please also see the revised manuscript for details. For easily reading, we use the TRACK function of MICROSOFT WORD. Please simply select “Accept changes” to get rid of the TRACK markers if you do not like the tracks. 

In addition, we would like to update our Funding Statement as follows:

This work was supported by the Supporting Project from Xinjiang Science and Technology Department (No.2020E0269).

Additional Editor Comments:

As evident from the Reviewer's comments, it is critical to re-address the images in Figure 1. It is very hard to see how the images provided come from serial sections, or how they correlate with each other, so that one can extract meaningful conclusions. Please provide a revised Figure 1 with appropriate images, and/or low magnifications of the images to allow for evaluation of location of panels and potential co-localizations, as well as the assessment of the same tumor.

Response: As suggested, we have further provided a revised Figure 2 (the original Figure 1) with high and low magnifications of the same tumor or normal sites. We have also made corresponding changes in the revised manuscript. Please check!

Reviewers' comments:

Reviewer #1: They have not essentially revised for my major concerns.

My biggest concerns are below. Although authors described that they use the serial section for IHC, but as far as I can see in the picture in Figure 1, I can not think that they have observed the same tumor or normal site in the continuous section. The photos should be shown the same tumor or normal site in the serial section.

Response: As suggested, we have further provided a revised Figure 2 (the original Figure 1) with high and low magnifications of the same tumor or normal sites. We have also made corresponding changes in the revised manuscript. Please check!

In figure4, the amounts of N-cadherin, Beta-catenin, and Vimentin proteins in these representative photos are not consistent with these quantitative graphs.

Response: As suggested, we have modified the representative and quantitative Western blot results in the revised Figure 4 to ensure that the amounts of N-cadherin, Beta-catenin, and Vimentin proteins in these representative photos were consistent with these quantitative graphs. We have also made corresponding changes in the revised manuscript. Please check!

Thus, I think that their experimental results are inadequate to draw their conclusions.

Response: As suggested, we have modified some results and some conclusion to ensure that the conclusions were supported by the experimental results. For example, the overstated conclusions have been modified. Please check the revised manuscript for details.

---

## [Decision Letter · Decision Letter 2]

18 Oct 2021

PONE-D-20-18825R2Correlation of breast cancer microcirculation construction with tumor stem cells (CSCs) and epithelial-mesenchymal transition (EMT) based on contrast-enhanced ultrasound (CEUS)PLOS ONE

Dear Dr. Ding,

Thank you for submitting your manuscript to PLOS ONE. After careful consideration, we feel that it has merit but does not fully meet PLOS ONE’s publication criteria as it currently stands. Therefore, we invite you to submit a revised version of the manuscript that addresses the points raised during the review process. Please fix the figure legends as indicated by the reviewers. Please submit your revised manuscript by Dec 02 2021 11:59PM. If you will need more time than this to complete your revisions, please reply to this message or contact the journal office at plosone@plos.org. Please include the following items when submitting your revised manuscript:A rebuttal letter that responds to each point raised by the academic editor and reviewer(s). You should upload this letter as a separate file labeled 'Response to Reviewers'.A marked-up copy of your manuscript that highlights changes made to the original version. You should upload this as a separate file labeled 'Revised Manuscript with Track Changes'.An unmarked version of your revised paper without tracked changes. You should upload this as a separate file labeled 'Manuscript'.If applicable, we recommend that you deposit your laboratory protocols in protocols.io to enhance the reproducibility of your results. Protocols.io assigns your protocol its own identifier (DOI) so that it can be cited independently in the future. For instructions see: https://journals.plos.org/plosone/s/submission-guidelines#loc-laboratory-protocols. Additionally, PLOS ONE offers an option for publishing peer-reviewed Lab Protocol articles, which describe protocols hosted on protocols.io. Read more information on sharing protocols at https://plos.org/protocols?utm_medium=editorial-email&utm_source=authorletters&utm_campaign=protocols.

We look forward to receiving your revised manuscript.

Kind regards,

Stella E. Tsirka

Academic Editor

PLOS ONE

Journal Requirements:

Reviewers' comments:

Reviewer's Responses to Questions

**Comments to the Author**

1. If the authors have adequately addressed your comments raised in a previous round of review and you feel that this manuscript is now acceptable for publication, you may indicate that here to bypass the “Comments to the Author” section, enter your conflict of interest statement in the “Confidential to Editor” section, and submit your "Accept" recommendation.

Reviewer #1: All comments have been addressed

Reviewer #2: (No Response)

2. Is the manuscript technically sound, and do the data support the conclusions?

Reviewer #1: Yes

Reviewer #2: Partly

3. Has the statistical analysis been performed appropriately and rigorously? 

Reviewer #1: Yes

Reviewer #2: I Don't Know

4. Have the authors made all data underlying the findings in their manuscript fully available?

Reviewer #1: Yes

Reviewer #2: Yes

5. Is the manuscript presented in an intelligible fashion and written in standard English?

Reviewer #1: Yes

Reviewer #2: No

6. Review Comments to the Author

Reviewer #1: I think it has improved. The IHC analysis result, which was a major concern, seems to be a mixture of those that can be compared at the same point in the continuous section and those that are not. However, since the tumor sample is very small, I understand that it is difficult to analyze. No more peer-reviewed rounds are needed.

Minor points.

1) The square part of the low-magnification photo of CD24 and the square part of the magnifying magnification are 90 degrees out of alignment. Please correct it as it is confusing.

2) The word Breast is missing in the graph on the upper right of Figure 4B.

3) There is a misspelling in the upper left corner of Table 1. The term of "cancer" is "caner".

4) The value of N-cadherin in Table 5 is 0.000. Please enter an appropriate value.

Reviewer #2: Figure legends are non-existence or very brief. Keys of various symbols and color codes should be explained in the legends to allow a better appreciation of the results and hence the conclusions. The relationship between figures 1 and 2 needs to clearly stated. It will be helpful to indicate n values for various analyses when possible.

7. PLOS authors have the option to publish the peer review history of their article (what does this mean?). If published, this will include your full peer review and any attached files.

Reviewer #1: No

Reviewer #2: No

---

## [Author Response · Author response to Decision Letter 2]

22 Nov 2021

Journal Requirements:

Response: We confirm that the reference list is complete and correct. No further changes to the reference list are needed. We did not cite any retracted papers. 

Reviewers' comments:

Reviewer's Responses to Questions

Reviewer #1: I think it has improved. The IHC analysis result, which was a major concern, seems to be a mixture of those that can be compared at the same point in the continuous section and those that are not. However, since the tumor sample is very small, I understand that it is difficult to analyze. No more peer-reviewed rounds are needed.

Minor points.

1) The square part of the low-magnification photo of CD24 and the square part of the magnifying magnification are 90 degrees out of alignment. Please correct it as it is confusing.

Response: Sorry for the confusion. We have corrected the low-magnification photo of CD24 in the revised Figure 2. Please check!

2) The word Breast is missing in the graph on the upper right of Figure 4B.

Response: Sorry for the typo. We have corrected this in the revised Figure 4. Please check!

3) There is a misspelling in the upper left corner of Table 1. The term of "cancer" is "caner".

Response: Sorry for the typo. We have corrected "caner" into "cancer" in the revised Table 1. Please check!

4) The value of N-cadherin in Table 5 is 0.000. Please enter an appropriate value.

Response: To be more clear, we have changed P=0.000 into P<0.001. Please check the revised Table 5 for details. 

Reviewer #2: Figure legends are non-existence or very brief. Keys of various symbols and color codes should be explained in the legends to allow a better appreciation of the results and hence the conclusions. The relationship between figures 1 and 2 needs to clearly stated. It will be helpful to indicate n values for various analyses when possible.

Response: As suggested, we have further provided more detailed information to the figure legends. Figure 1 shows the time intensity curve analysis of contrast-enhanced ultrasound (CEUS). The representative sections of CEUS were marked before surgery. Then, the tissue specimens were collected from these marked sections and subjected to immunohistochemical analysis of CSCs and EMT proteins (Figure 2). We have further clarified this in the revised manuscript. Please check! 

Additionally, the n values have been indicated in the figure legends and table titles. Please check!

---

## [Editor Report · Decision Letter 3]

25 Nov 2021

Correlation of breast cancer microcirculation construction with tumor stem cells (CSCs) and epithelial-mesenchymal transition (EMT) based on contrast-enhanced ultrasound (CEUS)

PONE-D-20-18825R3

Dear Dr. Ding,

We’re pleased to inform you that your manuscript has been judged scientifically suitable for publication and will be formally accepted for publication once it meets all outstanding technical requirements.

Kind regards,

Stella E. Tsirka

Academic Editor

PLOS ONE
---

## [Editor Report · Acceptance letter]

13 Dec 2021

PONE-D-20-18825R3 

Correlation of breast cancer microcirculation construction with tumor stem cells (CSCs) and epithelial-mesenchymal transition (EMT) based on contrast-enhanced ultrasound (CEUS) 

Dear Dr. Ding:

I'm pleased to inform you that your manuscript has been deemed suitable for publication in PLOS ONE. Congratulations! Your manuscript is now with our production department. 

Kind regards, 

on behalf of

Dr. Stella E. Tsirka 

Academic Editor

PLOS ONE